# Probing the spinor nature of electronic states in nanosize non-collinear magnets

Jeison A. Fischer[1,2], Leonid M. Sandratskii[1], Soo-Hyon Phark[1,3], Safia Ouazi[1], André A. Pasa[2], Dirk Sander[1] & Stuart S.P. Parkin[1]

Non-collinear magnetization textures provide a route to novel device concepts in spintronics. These applications require laterally confined non-collinear magnets (NCM). A crucial aspect for potential applications is how the spatial proximity between the NCM and vacuum or another material impacts the magnetization texture on the nanoscale. We focus on a prototypical exchange-driven NCM given by the helical spin order of bilayer Fe on Cu(111). Spin-polarized scanning tunnelling spectroscopy and density functional theory reveal a nanosize- and proximity-driven modification of the electronic and magnetic structure of the NCM in interfacial contact with a ferromagnet or with vacuum. An intriguing non-collinearity between the local magnetization in the sample and the electronic magnetization probed above its surface results. It is a direct consequence of the spinor nature of electronic states in NCM. Our findings provide a possible route for advanced control of nanoscale spin textures by confinement.

[1] Max-Planck-Institut für Mikrostrukturphysik, Weinberg 2, 06120 Halle, Germany. [2] Laboratório de Filmes Finos e Superfícies, Departamento de Física, Universidade Federal de Santa Catarina, 88040-900 Florianópolis, Brazil. [3] Center for Nanometrology, Korea Research Institute of Standards and Science, Daejeon 34113, Korea. Correspondence and requests for materials should be addressed to J.A.F. (email: jfischer@mpi-halle.mpg.de) or to L.M.S. (email: lsandr@mpi-halle.mpg.de) or to S.-H.P. (email: phark@kriss.re.kr).

Non-collinear magnets (NCM) exhibit a spatial variation of the magnetization direction, where helical[1,2] and skyrmionic[3–5] spin orders have lately attracted considerable interest. This interest is spurred by both, exploring the physical origin[6–8] of nanoscale NCM and applications in spintronics[9–16]. In view of current applications in spintronics typical lengths scales are in the nanometre regime, where NCM are embedded between other materials. The impact of the corresponding change of materials on the nanoscale and of the resulting symmetry breaking on the magnetization texture of the NCM is largely unexplored. It is a significant issue for both applications and fundamental physics.

The symmetry breaking resulting from lateral confinement modifies the electronic states of the NCM and consequently their magnetic order. A key aspect of NCM is that the electronic Hamiltonian does not commute with the operator of spin projection on any selected axis. This requires a representation of the electronic wave functions in the form of a two-component spinor

$$\psi(\mathbf{r}) = \begin{pmatrix} \psi_\uparrow(\mathbf{r}) \\ \psi_\downarrow(\mathbf{r}) \end{pmatrix}. \tag{1}$$

This description leads to electronic states that carry intrinsically a mix of spin channels. This spinor nature results in a non-trivial energy dependent non-collinearity between the atomic moments of the sample and the electronic magnetization at a given energy, as can be probed for example, by spin-polarized scanning tunnelling microscopy (sp-STM).

The present paper advances the understanding of nanoscale NCM by revealing the effect of lateral confinement on the physical properties of NCM. We discuss intriguing phenomena of general validity for a prototypical NCM, represented by a laterally confined Fe bilayer. The formation of NCMs in atomic layers has been shown to be governed by spin–orbit coupling through the Dzyaloshinskii–Moriya interaction[2,6,8,17,18] and/or by competing intralayer[7,19,20] and interlayer[21,22] exchange interactions. Theory predicted a helical spin structure for an infinitely extended Fe bilayer, and experimental results support this claim[7].

Here, we combine spin-polarized scanning tunnelling microscopy and spectroscopy (sp-STM/S) and *ab initio* based theory, supported by fundamental symmetry arguments, to unveil the underlying physics of the impact of spatial confinement on helical spin structures. We show that lateral nanoscale confinement induces significant modifications of the magnetic order of NCM in nanometre proximity to the interfaces. This novel finding has important consequences for potential applications of NCM in spintronics, where currently the robustness of the magnetization texture is tacitly assumed, and the impact of confinement and proximity is disregarded. A significant deviation away from collinearity between the local magnetization direction within the NCM sample and the electronic magnetization probed above its surface is also identified, and this sheds fresh light on the understanding of spin-polarization probed above a distorted NCM.

## Results

**Sample and electronic magnetization.** Figure 1 presents an introductory overview schematic on the physical background of our study. It contrasts the properties of an infinitely extended Fe bilayer, Fig. 1a, and a laterally confined bilayer, bounded by a ferromagnet and by vacuum, Fig. 1b. The magnetization direction of the infinitely extended Fe bilayer (orange) changes regularly with position. It gives rise to an ideal spin helix, where the regular spatial variation of orientation of the sample magnetization $\mathbf{m}_s$ is

shown by blue arrows in Fig. 1a. The electronic magnetization at a given energy, $\mathbf{m}_{el}(E)$ (red arrow), probed by sp-STM with a tip magnetization $\mathbf{m}_{tip}$ (black) above the sample is collinear with the magnetization direction in the sample at the position underneath the tip. The laterally confined system depicted in Fig. 1b shows a modification of the magnetic structure. The sample magnetization direction in proximity to the interfaces with a ferromagnet (blue) and vacuum (grey) deviates from that of the regular helical spin order. Note, that for the confined system of Fig. 1b the magnetization directions $\mathbf{m}_{el}(E)$ and $\mathbf{m}_s$ are non-collinear. The relative orientation of the two magnetization directions varies with energy of the probed electronic states.

We emphasize that both magnetizations ($\mathbf{m}_s$ and $\mathbf{m}_{el}$) are of electronic origin. Each electronic state contributes to the magnetization of the system. Summing over states at a given energy gives the energy-resolved magnetization. By integrating this magnetization up to Fermi level, we obtain the total magnetization. The sample magnetization ($\mathbf{m}_s$) reflects the magnetic structure of the material. The latter is commonly described in terms of directions of atomic moments. The atomic moments are the integrals of the total magnetization over atomic spheres. The main contribution to the atomic moments comes from localized 3d-states. On the other hand, the notation $\mathbf{m}_{el}(E)$ is used to refer to the energy-resolved magnetization extending to the vacuum and reaching the position of the tip.

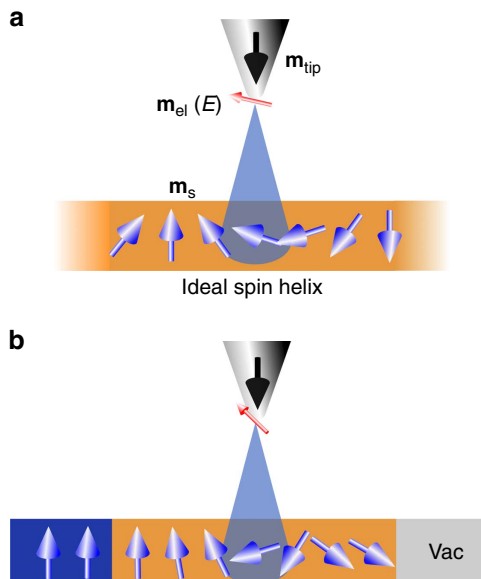

**Figure 1 | Schematic presentation of the effect of the lateral confinement on a NCM.** (**a**) Illustration of an ideal spin helix in a Fe bilayer sample (orange). Blue arrows depict the magnetization direction in the sample ($\mathbf{m}_s$) and the red arrow depicts the electronic magnetization ($\mathbf{m}_{el}(E)$) at a given energy $E$ in the vacuum region, as probed in sp-STM/S experiments. The black arrow shows the tip magnetization ($\mathbf{m}_{tip}$), which is fixed by the external magnetic field. The magnetization directions in the sample and above it are collinear. Note that for illustrative purposes we show parallel directions of the two magnetizations. Depending on the energy of probed electrons the directions can be both parallel and antiparallel. (**b**) Lateral interfaces with a ferromagnet (blue) and with vacuum (Vac) induce a modification of the magnetic structure of the Fe bilayer. The angular variation of magnetization directions between adjacent atoms is irregular and differs at each position. In contrast to the case of the ideal helix in **a** the direction of electronic magnetization probed above the sample is non-collinear with respect to the atomic moment underneath, within the sample.

This part of the magnetization is mainly due to s-states, which are spin polarized due to the interaction with the atomic 3d-states.

**Magnetic structure distortions caused by lateral confinement.** Here, we present sp-STM/S measurements on NCM's, namely, bilayer Fe nanoislands with topmost Fe in bridge positions on Cu(111)[7,23,24]. The details on tip and sample preparation are given in the Methods section. The signal in sp-STM/S measurements depends on the angle $\theta$ between tip magnetization $\mathbf{m}_{tip}$ and sample electronic magnetization $\mathbf{m}_{el}$ probed at the tip position. This gives rise to a contribution to the STM signal in proportion to the projection of the electronic magnetization on the direction of the tip magnetization $\mathbf{m}_{tip} \cdot \mathbf{m}_{el} = m_{tip}m_{el} \cos(\theta)$ (ref. 25). In the experiment, the direction of the tip magnetization is normal to the sample surface throughout the energy range probed in our experiment (Supplementary Fig. 1; Supplementary Note 1), and this direction serves as the global reference axis, chosen as the z-axis.

The intrinsic magnetic structure of an Fe bilayer is a prototypical realization of an exchange-driven spin helix[7]. To explore the spatial dependence of the spin magnetization of a nanosized confined Fe bilayer we measure differential conductance asymmetry $A_{dI/dV}$ (ref. 26). For the Fe|Co island, we use the ferromagnetic Co as a reference. The asymmetry follows as:

$$A_{dI/dV}^{Fe|Co} = \frac{dI/dV_{AP} - dI/dV_{P}}{dI/dV_{AP} + dI/dV_{P}}, \qquad (2)$$

where $dI/dV_{AP}$ and $dI/dV_{P}$ are the $dI/dV$ signals measured with antiparallel (AP) and parallel (P) magnetization configurations of the tip and sample reference (Co), respectively. The magnetic field values were chosen from the $dI/dV$ hysteresis curve (Supplementary Fig. 2; Supplementary Note 2) to obtain the appropriate AP and P magnetization configurations. In the case of the pure Fe island, we compare a measurements in-field, which reveal the magnetic structure in the sample, with measurements without field, which gives no contrast related to the magnetic structure[7]. Here, the $dI/dV$ asymmetry is defined by

$$A_{dI/dV}^{Pure\,Fe} = \frac{dI/dV_{In\,field} - dI/dV_{No\,field}}{dI/dV_{No\,field}}. \qquad (3)$$

where $dI/dV_{In-field}$ and $dI/dV_{Nofield}$ are the $dI/dV$ signals measured with and without magnetic field, respectively. On the basis of the generalized Tersoff-Hamann model[25–27], we can link the $dI/dV$ asymmetry $A_{dI/dV}$ to the spin polarization of the tip, $P_T$, and of the sample at the tip apex position projected on the global axis, $P_S$:

$$A_{dI/dV} = -P_T P_S, \qquad (4)$$

as discussed before[26]. $P_S$ is closely related to $m_{el,z}$. We obtain asymmetry maps for two types of samples, a bilayer Fe island of 10 nm width and a narrower bilayer Fe stripe of 3–4 nm width, surrounding a ferromagnet bilayer Co core. We refer to the latter as Fe|Co island for short.

Figure 2a–e present experimental results on the Fe island: Fig. 2a constant-current STM image of the Fe island, Fig. 2b differential conductance ($dI/dV$) asymmetry map of the Fe island, Fig. 2c energy-resolved differential conductance spectra, Fig. 2d line scans at different energies through the differential conductance map of Fig. 2c,e energy profiles of the differential conductance at different positions. Figure 2f–j present experimental results on the Fe|Co island: Fig. 2f constant-current STM image of the Fe|Co island, Fig. 2g differential conductance asymmetry map of the Fe|Co island, Fig. 2h energy-resolved differential conductance spectra, Fig. 2i line scans at different energies through the differential conductance map of Fig. 2h,j energy profiles of the differential conductance at

different positions. The boundary between Fe and Co core is obtained by spatially-resolved differential conductance $dI/dV$ maps, see Supplementary Fig. 3 and Supplementary Note 3.

Figure 2b,g shows the measured differential conductance asymmetry for both samples at an energy of $-0.65$ eV (Supplementary Fig. 4; Supplementary Note 4). The labelled black lines in Fig. 2b,g identify line scans along which the energy dependence of the asymmetry was measured. The results of the position and energy resolved measurements are given in Fig. 2c,h. A visual comparison of Fig. 2c,h shows dramatic differences in the differential conductance asymmetry of the two samples. Line scans through the energy dependent $dI/dV$ asymmetry data of Fig. 2c shown in Fig. 2d reveal a spatial variation of the signal, which can be rather well fitted by a sine-function. This identifies the NCM in the inner part of the larger Fe bilayer sample of Fig. 2a as an experimental approximation of an ideal spin helix. The similarity of this NCM to an ideal spin helix is further corroborated by the analysis of the energy dependence of the asymmetry signal for different spatial positions within the Fe bilayer, Fig. 2e. We observe that all curves exhibit a similar energy dependence. This is a symmetry-determined feature providing experimental evidence for the generalized translational symmetry, which characterizes the ideal helix.

In order to study the influence of the interfaces Co–Fe and Fe-vacuum on the NCM of Fe, we measure the energy dependence of the $dI/dV$ asymmetry signal along lines from the Co regions through Fe towards the surrounding Cu substrate, labelled 2 and 3 in Fig. 2g. Figure 2h presents the data for the line scan along direction 2. Data for the other direction are given in Supplementary Fig. 5 and Supplementary Note 5. In striking contrast to the results of Fig. 2c, we observe a lack of regularity in the $dI/dV$ asymmetry signal within the Fe rim. The irregular nature of the $dI/dV$ asymmetry variation is also obvious from the large deviation between the experimental data and the sine-functions, shown by the dotted grey lines in Fig. 2i. The position-resolved energy dependence of the $dI/dV$ asymmetry in Fig. 2j shows very different shapes of curves for different spatial positions, in contrast to the similar curve shapes for an almost ideal helix, Fig. 2e. For all energies we find that the amplitude of the spatial variation of the $dI/dV$ signal fades out within the Fe region towards its boundary with vacuum.

These experimental data suggest a strong distortion of the intrinsic helical spin structure of the Fe bilayer in proximity to interfaces with Co and vacuum. A proper appreciation of the results requires an in-depth understanding of the contrast mechanism in sp-STM. The question arises, how the electronic magnetization $\mathbf{m}_{el}$, as probed by the tip, is related to the atomic magnetization of the sample. For ferromagnetic and ideal spin helix structures, both magnetization directions are collinear, as required by symmetry. However, in confined NCM this collinearity is lost, as will be elaborated in conjunction with theory in the next sections. Only the combined experimental and theoretical treatment unravels the distorted spin structure of the system.

**Effect of nanosize lateral confinement on NCM.** We apply density functional theory (DFT) generalized to the case of NCM (see refs 28–31) to study the effect of nanoscale lateral confinement on the magnetic structure and electronic properties of bilayer Fe. To approximate the experimental situation we treat infinite Fe bilayers and an Fe bilayer stripe confined between Co and vacuum. The Fe atomic sites are arranged for the calculations to mimic the epitaxial order on Cu(111), with topmost Fe in bridge-site stacking[7]. The calculations provide the self-consistent magnetic structure of the system where the directions of the spin moments in different atomic spheres can be arbitrary. We remark that in Fe

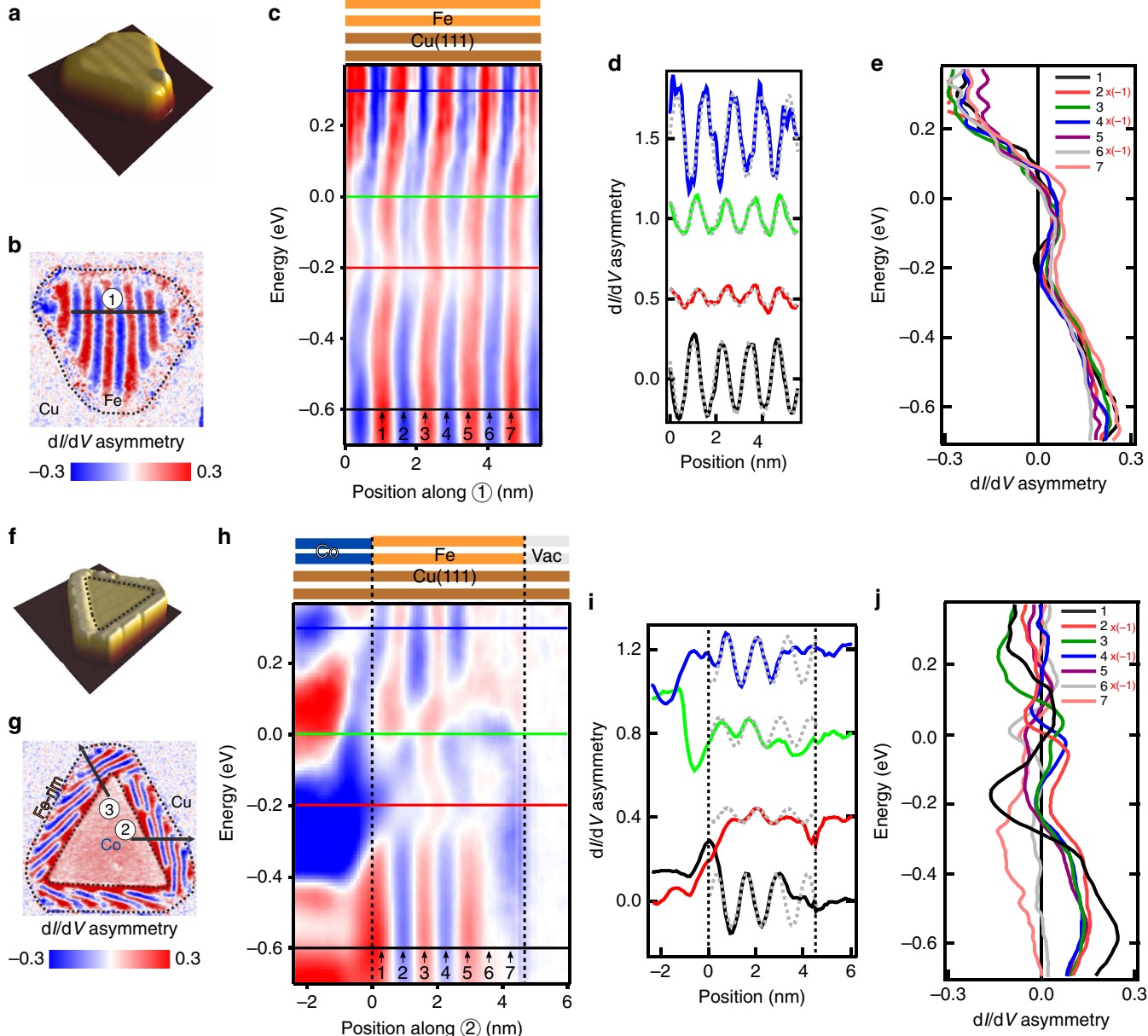

**Figure 2 | Almost ideal and distorted helical spin structures in confined Fe bilayers.** (**a,f**) Perspective view of constant-current-STM images of a pure Fe bilayer island (**a**) (1 nA, −0.6 V, 10 × 10 nm) and of a bilayer Fe-rim around a bilayer Co island (**f**) (1 nA, 0.5 V, 24 × 24 nm) on Cu(111). Dotted lines indicate the boundaries. (**b,g**) Maps of the d$I$/d$V$ asymmetry of islands **a,f** respectively. The d$I$/d$V$ asymmetry maps are calculated from two d$I$/d$V$ maps in different magnetic configurations. Measurements conditions: $V_b$ = − 0.65 V, $I_{set}$ = 1 nA, $\mu_0 H$ = − 1.0 T for **b**; $V_b$ = − 0.65 V, $I_{set}$ = 1 nA, $\mu_0 H$ = − 1.2 T for **g**. The outer Fe-rim is defined by half-height in constant-current images. (**c,h**) Energy resolved d$I$/d$V$ asymmetry extracted from line scans along direction 1 in **b**, and along direction 2 in **g**, respectively. (**d,i**) d$I$/d$V$ asymmetry line profiles (continuous line) at different energies, as defined by the coloured horizontal lines in **c,h** respectively (energies: black curve: − 0.6, red: − 0.2, green: 0.0, and blue: + 0.3 eV). A sine-function (dashed grey line), representing an ideal spin helix, is shown for comparison. Line profiles are shifted vertically for clarity by 0.5 and 0.4 eV for **d,i** respectively. (**e,j**) Energy resolved d$I$/d$V$ asymmetry at fixed positions 1…7 in the Fe regions. The d$I$/d$V$ asymmetry curves in **a,b** were extracted at the positions 1…7 identified at the bottom of **c,h** respectively. The positions correspond to the maxima and minima of the asymmetry signal at − 0.6 eV. These extrema are separated by 0.64 nm, corresponding to half wave length of the spatial asymmetry modulation of the ideal spin helix. The curves obtained for the minima were multiplied by − 1 to account for the antiparallel orientation of the magnetization of the tip and the local magnetization in the sample.

the spin-orbit coupling (SOC) is much weaker than the exchange interaction, and it is neglected in our calculations. Further details of the calculations are given in Methods and in Supplementary Fig. 6 and Supplementary Note 6.

The magnetic structure of an infinite Fe bilayer is an exchange-driven spin helix[7] (see Supplementary Fig. 6; Supplementary Note 6). For the understanding of the physical effect of the lateral confinement it is important to realize that the very existence of

the helical spin structure is intimately connected with the translational symmetry of the system. Since the lateral confinement of the Fe film breaks the translational symmetry, all regularities in the system that are the consequences of this symmetry are disturbed. This renders the Fe atoms inequivalent and distorts the spin structure.

In Fig. 3, we present the results of the calculations for an infinite bilayer Fe and for the system 16 Co/30 Fe/8 Vac, where

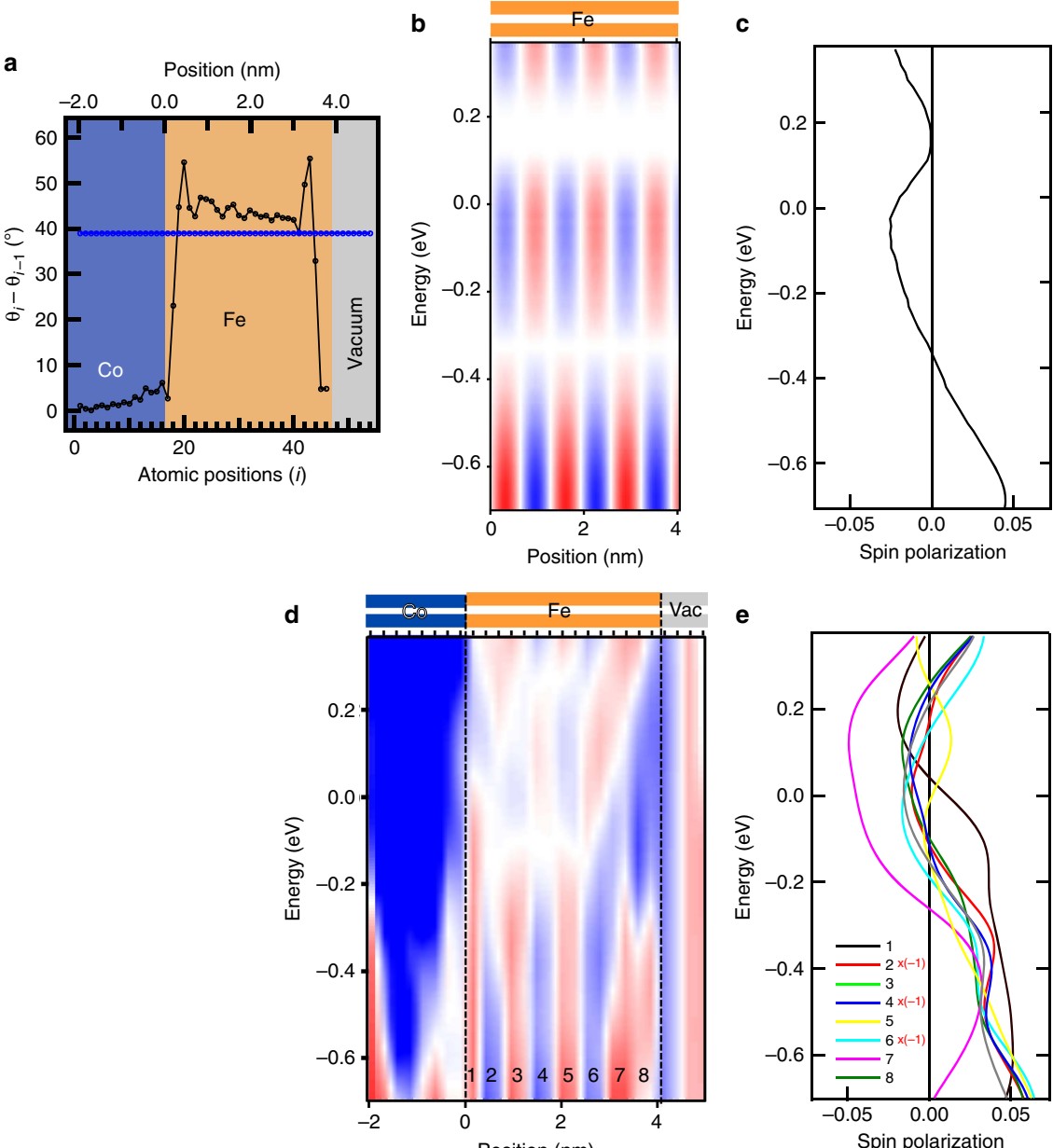

**Figure 3 | Calculated spin-polarization of NCMS in Fe bilayers.** (**a**) Density functional theory based calculation of the angle between the atomic moments of neighbouring atoms in an ideal spin helix (blue data points) and along a 16 Co (blue)/30 Fe (orange)/8 Vac (grey) bilayer stripe (black data points) (**b,d**) Calculated spatial and energy dependence of the electronic spin magnetization projected onto a global axis (sample normal) above the bilayer Fe film. Two samples are analysed: an infinite Fe bilayer (**b**), and a 16 Co/30 Fe/8 Vac bilayer stripe (**d**). Red, white and blue represent positive, zero and negative spin polarization, respectively. (**c**) Energy-resolved projection of the electronic magnetization on the direction of the corresponding atomic moment in the sample for the case of ideal helix, calculated for a vacuum sphere with centre 4 Å above the center of the corresponding Fe atom. (**e**) Energy-resolved projections of the electronic magnetization on the directions of the corresponding atomic moments for eight spatial Fe positions in the confined sample 16 Co/30 Fe/8 Vac, calculated for vacuum spheres 4 Å above the surface. The selected lateral positions are identified in **d**.

the numbers give the number of atomic spheres in a bilayer arrangement. The calculated variation of the angle between neighbouring magnetic moments is given in Fig. 3a. The angles calculated for the Co/Fe/Vac system (black data points) deviate pronouncedly from the constant angle of the ideal spin helix, given by the blue points in Fig. 3a. The largest differences with respect to the ideal spin helix are obtained close to the interfaces. The interfaces distort the magnetic structure differently. For example, the angle between magnetic moments of first and second Fe-atoms next to Co is roughly half of that of the ideal spin helix. However, near the interface with vacuum an almost

parallel alignment of adjacent Fe magnetic moments is calculated. Also the calculated magnetic moments and electronic charges show pronounced deviations from that of the ideal spin helix (Supplementary Fig. 7; Supplementary Note 7). In addition, we calculate the magnetic structure of bilayer Fe with symmetric interfaces Vac/Fe/Vac and Co/Fe/Co. We find a further previously undisclosed nanosize effect: the magnetic structure in the central part of the Fe stripe depends on the width of the stripe (Supplementary Fig. 8; Supplementary Note 8).

Figure 3b–e contrast the properties of the electronic magnetization above the sample for the infinite Fe bilayer

(Fig. 3b,c) and the laterally confined sample (Fig. 3d,e). Figure 3c shows the energy dependence of the projection of the spin magnetization density on the direction of the corresponding atomic moment. This quantity is identical for all Fe atoms of the spin helix. Figure 3b shows the energy and position resolved projection of the spin-magnetization density on the global axis. The position dependence of the global projection is obtained from the local quantity by taking into account the regularity in the directions of the atomic moments of the helix.

Next we focus on the 16 Co/30 Fe/8 Vac system, where we have identified a strong distortion of the magnetic structure with respect to the ideal spin helix. Figure 3d presents the result of the calculation of the energy- and position-resolved projection of the electronic magnetization on the global axis. The plots reveal a drastic difference compared to the corresponding data for the ideal helix, Fig. 3b. There is a pronounced and irregular variation of the spatial dependence of the data with energy. The translational symmetry of the data shown in Fig. 3b for the ideal spin helix is lost for the confined sample described in Fig. 3d. We conclude that, in agreement with the study of the magnetic structure distortions (Fig. 3a), the interfaces with Co and vacuum affect the electronic magnetization differently. The comparison between the experimental results of Fig. 2h and theory in Fig. 3d shows a qualitative agreement, revealing a heavily distorted modulation of the magnetization.

In Fig. 3e we present the calculated energy dependence of the electronic magnetization at positions 1…8, as indicated in Fig. 3d. The calculations reveal a pronounced difference in electronic spin magnetization at different spatial positions. This reflects the different electronic properties of Fe atoms at different spatial positions in the laterally confined sample. This contrasts with the case of the ideal helix in Fig. 3c. This calculated similarity (difference) between the spatially-resolved electronic properties of Fe for the infinite (confined) system compares convincingly with the experimental data presented in Fig. 2e,j.

Thus, we demonstrate that distortions of the NCMS in comparison with an ideal spin helix are a direct consequence of nanosize and proximity effects caused by the presence of lateral interfaces.

**Spinor nature of the electronic states in NCM**. The complexity of the data presented in Figs 2h and 3d indicates the involved nature of the energy dependence of the magnetization probed above the surface of a laterally confined NCM. To extract quantitative information from the differential conductance asymmetry maps of Fig. 2h, we fit the experimental data at each energy $E_i$ with a function $A_i \cos(kx + \varphi_i)$. Here, $kx + \varphi_i$, is regarded as the angle $\theta$ between the magnetization direction of the tip $\mathbf{m}_{tip}$ and the electronic magnetization $\mathbf{m}_{el}$. Further details about the fitting are given in Supplementary Note 9. We restrict this fit to a spatial region within 3 nm to the Co boundary, where a regularity in the variation of the data is observed. The resulting fit shown in Fig. 4a is in a decent agreement with the respective region of Fig. 2h.

Figure 4b shows the phase $\varphi$, obtained from the fit, as a function of energy. Here, the important new aspect is the continuous variation of the direction of the electronic magnetization with energy at a given spatial position. This is seen as a gradual change of $\varphi$ between 0 and $\pi$. The continuous variation of the direction of the electronic magnetization above the sample reveals its non-collinearity with the corresponding atomic magnetization. Note that the magnetization of the sample is non-collinear, but here we refer to the non-collinearity between the magnetic moment ($\mathbf{m}_s$) and the electronic magnetization ($\mathbf{m}_{el}(E)$), as indicated in the sketch of Fig. 1b.

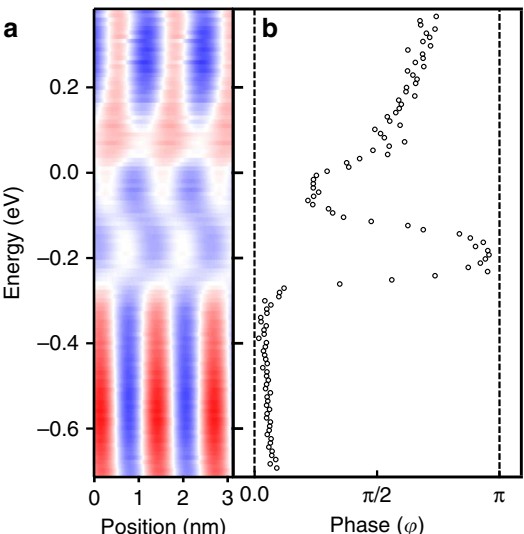

**Figure 4 | Phase shift of the magnetization measured above the Co|Fe system.** (**a**) Resulting fit of the d$I$/d$V$ asymmetry (Fig. 2h) of the inner Fe region within 3 nm of the Co interface, as described in the text. Red, white and blue indicate positive, zero and negative asymmetry, respectively. (**b**) Energy dependence of the phase angle ($\varphi$) used for the fit in **a**. Dashed lines indicate phase angles of zero an $\pi$, which correspond to parallel and antiparallel orientations between the atomic magnetic moment direction within the Fe bilayer and that conveyed by the electronic states measured 4 Å above the surface with sp-STM, respectively.

The explanation of the observed energy-dependent non-collinearity between the probed electronic magnetization and the atomic moments of the sample relies on the spinor nature of the electronic wave functions. In Fig. 5 we show the calculated energy dependence of the angle between electronic magnetization and atomic moment for the ideal helix and for a representative Fe atom in the Co/Fe/Vac sample. For the reasons explained below we refer to this angle as spinor angle. In the case of the ideal spin helix (blue data points), the electronic magnetization is, at any energy, collinear with the atomic moment, resulting in a spinor angle of either zero or $\pi$ (see Fig. 5). Variations between these two values occur in a discrete, step-like manner. In contrast, for the Co/Fe/Vac system the spinor angle shows a smooth energy dependence, as revealed by the black data points in Fig. 5. The sketches in Fig. 5 show examples of the deviation of the electronic magnetization from the atomic moment by 20° and 160° at −0.2 and −0.1 eV, respectively.

**Discussion**

To understand the physics of the effect we, first, consider the case of a ferromagnet. Here all atomic moments are parallel to each other and constitute a natural direction of the spin quantization axis. Since the Hamiltonian of a ferromagnet commutes with the operator of spin projection on the magnetization direction, each of the electronic states is characterized by a certain spin projection, which serves as a good quantum number of the electronic states. Therefore, the contribution of an electronic state to the spin magnetization can be either parallel or antiparallel to the magnetization direction. We note that in collinear systems with strong spin-orbit coupling, the non-collinear part of the magnetization density has been discussed previously[32]. In these studies the non-collinearity originates from spin mixing driven by spin-orbit coupling. Bode et al.[33] consider a special type of the intra-atomic non-collinearity in the ferromagnetic system caused

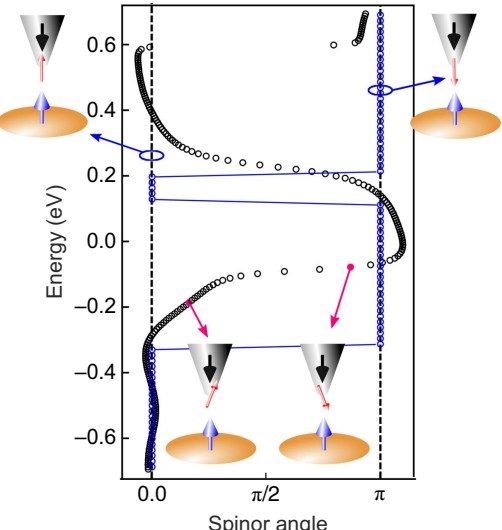

**Figure 5 | Calculated energy dependence of the spinor angle.** The blue data points represent the calculated energy dependence of the spinor angle for an ideal spin helix. The angle is either zero or π, and variations occur in a step-like manner. Calculated energy dependence of the spinor angle of the 16 Co/30 Fe/8 Vac system. The spinor angle is defined as the angle between the electronic magnetization direction calculated 4 Å above the surface (as probed in sp-STM/S experiments) and the atomic moment direction in the Fe atom in the sample surface. The continuous variation of the spinor angle with energy is the hallmark of the non-collinearity between the electronic magnetization and the atomic moment. The small sketches visualize the variation of the spinor angle with energy.

by the different directions of the spins of different atomic 3d-states. In our study, we do not see any sign of this effect and do not consider it. The energy dependence of the direction of the magnetization discussed by us is the result of inter-atomic processes in distorted non-collinear magnetic configurations.

In a NCM, the electronic wave functions have the form of a two-component spinor, see equation (1), where both spin components are non-zero. The electronic states described by such a spinor necessarily have a non-zero component of the spin magnetization transversal to the direction of the quantization axis (Supplementary Note 10). However, the symmetry of the ideal spin helix leads to a degeneracy of electronic states with symmetry-related wave functions, whose transversal magnetization components, at any energy, compensate each other. The result is that the electronic magnetization in the vacuum region above the sample is collinear to the corresponding atomic moment.

In the case of a laterally confined sample the symmetry-determined compensation of the transversal spin-magnetization components does not take place. As a result, the direction of the spin magnetization is not a symmetry-determined property. Consequently, the spinor nature of the electronic states, leading to the energy-dependent non-collinearity of the electronic magnetization and atomic moment, is experimentally accessible. The qualitative similarity between the energy dependent phase φ (Fig. 4b) and the spinor angle (Fig. 5) corroborates this result.

The Fe sample in Fig. 2a has been treated as an approximation of an ideal spin helix. However, a detailed inspection of Fig. 2c reveals noticeable deviations from a strictly vertical contrast, as calculated for an ideal spin helix, Fig. 3b. These deviations can be ascribed to the influence of lateral confinement, induced by the finite width of the Fe bilayer, as shown in the Supplementary Figs 9 and 10 and discussed in the Supplementary Note 11. Consequently, the distortions also reveal the spinor nature of the

electronic states in this sample. This points to the importance of energy-dependent studies of the d$I$/d$V$ signals in sp-STM/S experiments to probe the spinor physics in laterally confined NCM.

In conclusion, we combine sp-STM/S experiments and first-principles calculations to study the physical effects related to nanoscale lateral confinement of helical NCM. These phenomena of general validity are exemplified for the case of bilayer Fe, which is a prototypical helical NCM. We demonstrate in experiment and theory that lateral interfaces results in a non-uniform distortion of the intrinsic magnetic structure. This distortion of the magnetic order is intimately linked to the loss of translational symmetry of confined systems. It leads to the in-equivalence of atoms within the NCM. Size- and interface-dependent distortions in the magnetic structure of the NCM result, and they are innately connected with the modification of the electronic structure. We identify the novel effect of the non-collinearity between sample and electronic magnetizations, which occurs for laterally confined NCM. lt is absent for collinear magnets and ideal spin helix structures. This phenomenon is a direct consequence of the spinor nature of the electronic states in NCM. Thus, the spinor nature of electronic states is observable in sp-STM/S experiments. The phenomenon of non-collinearity between electronic magnetization and atomic moments, elaborated in the paper, is also relevant for non-adiabatic spin transfer torque processes[12,34,35], which are envisaged for efficiently dealing with data in modern racetrack memory devices[10,36–38]. Our work opens the way to exploit the effects of lateral confinement for an advanced control and tuning of spin and electronic structures on the nm scale, which is compatible with nanotechnology devices.

## Methods

**Experiments.** The experiments were performed in an ultra-high vacuum (UHV) chamber (base pressure $<1 \times 10^{-11}$ mbar) equipped with an Omicron scanning tunnelling microscope operating at 10 K and a superconducting magnet for magnetic field of up to 7 T, normal to the sample surface. The Cu(111) single crystal substrate (MaTeck GmbH) was cleaned by cycles of $Ar^+$-sputtering (1 keV, 1.2 μA sample current, 15 min per cycle) and subsequent heating at 700 K for 15 min until defect-free, atomically flat and clean, and large ($>200$ nm) terraces are observed in STM. We deposit first 0.24 monolayer (ML) Co, then 0.28 ML Fe on the cleaned Cu surface at 300 K in UHV. We used Cr/Co-coated W and electro-chemically etched Cr tips in our sp-STM experiments[39,40]. Details of the sample and tip preparations are described in refs 7,41,42. STS spectra were measured by a lock-in technique with a modulation bias voltage of a root-mean square amplitude of 20 mV at a frequency $v = 4$ kHz. The tunnel current $I(V)$ and the differential conductance d$I$/d$V$ were detected simultaneously. For a simultaneous measurement of constant current maps and differential conductance maps, we obtain STS spectra at each position of a scan area. To obtain an atomic scale spatial resolution ($<2$ Å) of a differential conductance map, we chose a 150 × 150 grid size for a 25 × 25 nm image. The measurement time was 12–24 h for this spectroscopic mapping.

**Theory.** The DFT calculations are performed with the augmented spherical waves method generalized to the case of arbitrary non-collinear magnetic configurations as described in, for example, ref. 28. We employed the exchange correlation functional in the form suggested by von Barth and Hedin[43]. The space is filled with atomic spheres and the direction of the spin magnetization in each sphere is self-consistently determined. We use a super cell geometry where the vacuum region is described by empty spheres. The spin-polarization of the electronic states in the vacuum region was consequently taken into account. For comparison with the experimental asymmetry signal we considered the vector spin magnetization calculated for the spheres with the centre lying 4 Å above the surface. In the calculations for incommensurate helical spin structures of infinite Fe bilayer we use the generalized Bloch theorem that reduces the lateral size of the unit cell to the size of the chemical one. In vertical direction the super cell contains two Fe layers and four layers of empty spheres. For laterally confined bilayers we use super cells with large lateral size and replace part of the Fe atoms with Co or empty spheres depending on the character of the confining interfaces. The calculated electronic wave functions have the form of two-component spinors. Depending on the concrete physical quantity we are interested in, the spinors can be written with respect to the local atomic or global spin quantization axes. The spin moment of each electronic state is calculated as a three-dimensional vector. This allows to

reveal the energy dependence of the direction of the electronic magnetization. For comparison with experiment the calculated energy dependencies of the magnetization densities are smoothed with the Lorentzian function with characteristic width of 0.16 eV. More details on the calculation approach are given in Supplementary Note 6.

**Data availability.** All relevant data are available from the authors on request.

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

## Acknowledgements

We acknowledge Dr M. Corbetta for fruitful discussions and W. Greie for technical support. The work was partially supported by Deutsche Forschungsgemeinschaft grant SFB 762.

## Author contributions

J.A.F, S.H.P. and D.S. conceived the experiment, J.A.F and S.H.P performed the sp-STM/S measurements, L.M.S. performed and analysed the DFT calculations. All authors discussed the results and contributed to the writing of the paper.

## Additional information

**Competing financial interests:** The authors declare no competing financial interests.

