## [Peer Review File · Nature Communications]

Reviewers' comments:

Reviewer #1 (Remarks to the Author):

The authors study the influence of nanoscale lateral confinement on the properties of a non-collinear ferromagnet by spin-polarized STM/STS.

They establish that at all tunneling biases the differential conductance asymmetry at different positions along the helix is determined by a unique phase in systems without boundaries. This is to be expected as locally the spins from different bands are expected to be collinear - even if there is no proof of such a statement and certainly systems might be envisaged where this is different.

In line 87/88 the authors dub this expectation "generalized translation symmetry", but I am not sure if introducing this terminology really helps the reader in understanding the underlying physics. This remark goes for more of the terminology used/introduced by the authors - for instance the authors distinguish between electronic, atomic and spin magnetization, which is not very useful nor physical, as all magnetization discussed in the paper has the same source: the electron spin. Also the authors introduce for instance "atomic spheres" which are a useful concept in band-structure methodology, but barely a physical quantity. Another one is the use of "spinor" and "spinor nature" of electronic states. Spin-states are of spinor nature - little controversy about that. The spinor nature of *bands* is *not* probed in the present work (it can be in for instance topological insulators). Apart from this there is no experiment that can avoid probing the spinor nature of magnetization. These kind of embellishments are imprecise and in the end obscure the findings of the paper - which are really interesting.

The main observation of the authors is that the interaction with boundaries - be it magnetic or non-magnetic - change the energy dependence of the differential conductance asymmetry. This I find an intriguing result.

It should be mentioned that the authors do not define the "differential conductance asymmetry" in the main text and only do so in the Methods section and elaborate up on it in the Supplementary Information. The authors connect this asymmetry directly to "spin polarization", which in Fig. 3 is provided in arbitrary units. Again, introducing this kind of terminology and treating it in a vague manner does not help the clarity of the paper. Rather I would expect a discussion in the paper how and under which conditions the energy dependence of the "differential conductance asymmetry" is related to the energy dependent real-space spin-polarization of the electronic bands.

The observations of the authors are compared to the result of band structure calculations. This works well. The interesting conclusion of the authors is that - paraphrasing in my own words - close to the boundaries the local magnetization is non-collinear. This is concluded on basis of the energy dependence of the conductance asymmetry close to the boundary: rather than a change of amplitude (which would imply an energy dependence of a collinear longitudinal polarization), there is an energy dependent change of *phase* of the helix - even if these two alternative interpretations are not directly compared in the paper. (What worries me is that the pictures of e.g. Fig. 2 can be distorted by the asymmetry being high at energies at which the tunneling density of states is very low - rather one would prefer a measure of the asymmetry at which the DOS is high.)

This is main observation conceptually interesting, and most likely caused by scattering states at the boundary or ferromagnetic states leaking into the NCM from the neighboring ferromagnet causing an energy dependent frustration of the local magnetization. Reporting it is definitely of interest to the broad readership of Nature Communications in my opinion. However, the manuscript would profit substantially from a more straightforward formulation of the observation, a more critical analysis in one or two places and a thorough discussion on the physical cause of the "local non-collinearity" of the magnetization. Such would be required to make the manuscript fit

for Nature Communications.

Reviewer #2 (Remarks to the Author):

The manuscript of Fisher and coworkers entitled ‘‘ Probing the spinor nature of electronic states in nano size non-collinear magnets ’ ’ describes spin-resolved scanning tunneling conductance measurements and density-functional theory calculations of a nanoscale magnetic sample consisting of either isolated Fe double layer islands or narrow Fe double layer stripes grown around Co island cores. The main result emphasized by the authors is the fact that the actual magnetic quantization axes and the electronic magnetization in non-collinear magnets do not necessarily coincide but may differ by surprisingly large angles. Although some aspects of the physics have been discussed previously and are somewhat confusingly presented in the manuscript under review here (as I will point out below) the systematics applied in both experiment and theory lift the study by Fisher and co-workers to a new level not achieved previously.

The data are certainly relevant for scientists working in the field of magnetism, but may also find interest in the broader solid-state physics community. Overall, the manuscript is written in a very comprehensible style. The authors managed to well organize the most relevant data within the manuscript and describe less important but informative data in the supplemental part. As far as I can judge, the experimental data have been obtained on a suitable sample system, are of very high quality, and have been analyzed carefully to extract the relevant information.

Most claims are substantiated by both experimental data and direct comparison with theory. What I didn't like was the explanation of how the spinor angle ϕ (presented in Fig. 4 and 5) is obtained. To my opinion the procedure is not properly discussed. To fix this point the authors should consider adding another part to the supplemental material where this fitting procedure is described in detail. Furthermore, it is not clearly addressed what the role of the confinement is. Is it the fact that inversion symmetry is broken or is there some electronic interference effect?

Another issue that appears potentially confusing to me is the following: In the main part of their manuscript the authors seem to suggest that in collinear magnets or infinite films the electronic magnetization is always aligned with the magnetic quantization axes. However, the non-collinear part of the magnetization density (in collinear systems) has been discussed previously (see, for example, PRL 76, 4420 and Refs. 2-5 therein) and also verified experimentally (PRL 86, 2142). In these studies (some of which shall be cited to properly discuss the scientific background) the non-collinearity originates from band structure effects or from spin mixture driven by spin-orbit coupling.

One (probably minor) point:

Why is the experimentally observed asymmetry (Fig. 1e) so much larger than the calculated spin polarization (Fig. 2c)? Usually it's the other way around.

Reviewer #3 (Remarks to the Author):

This manuscript reports on a surprising change in the non-collinear phase of helical Fe structures as detected using spin-polarized dI/dV maps. The effect appeared only on Fe regions in contact to Co islands, on the one side, and was absent on pure Fe islands. The experimental results are of high quality, and the authors provide a very complete analysis. DFT simulations find that the Co-Fe-Vacuum slab shows variation in the spin magnetization with energy in a similar range.

I find, however, that this manuscript leaves important issues unclear to this reader regarding to the physical origin of the reported effect. I will try to express them in the most succinct way:

Fig. 4 represents the main result of this manuscript: the energy dependence of the non-collinear phase difference between atomic neighbours measured for the Co-Fe-Vacuum case. The spectacular observation of a phase variation in the range between -0.2 eV to 0 eV is correlated to the calculated spinor angle, which shows also variations between -0.2 eV and +0.2 eV. This apparently confirms that the measurements reproduce the spinor nature of the electronic states (as they mention in page 202). However, I never had any doubt of that the sp-STs would reproduce variation in the electronic spin polarization; this is the basis of sp-STs (as said also in line 44).

The paper leaves still unsolved the origin of the main claim, namely, that the asymmetric barrier case shows such peculiar energy evolution of the NCM phase. Many questions regarding the origin of such phase change, the energy range where it occurs, and the role of the Co interface remain open.

The authors mention in the abstract that "theory reveals that the lateral confinement-induced symmetry breaking is the driving force..." , but this conclusive sentence is not supported in the presented results neither in the discussion. Why confinement produces this effect? I also miss more specific details about the theory employed, that would allow a reader to reproduce the results: which code/functional do the authors use? does theory take into account spin polarization of states several Angstroms above the surface? If so, at which distance?

Additionally, pure Fe islands also show variations of NC spin polarization in dI/dV profiles, shown as supplementary Material for the case of pure Fe islands. However, although DFT simulated the vacuum-Fe-vacuum case (also in Sup. material), the authors did not present the calculated energy dependent of spinor components. Can they provide an explanation of this effect on pure Fe islands too?

Although the authors present results of incredibly high quality, both theoretical and experimental, the main claim of the manuscript origin of the main observation, non-collinearity between sample and electronic magnetization, is not fully explained. It is also claimed that "It is absent for collinear magnets and ideal spin helix structures." But for pure Fe islands, it is also observed to less extend. I suggest the authors to work out more in detail this part of the manuscript, without which I cannot recommend publication of this manuscript in its current form.

Reviewers' comments:

Reviewer #1 (Remarks to the Author):

We are glad to read that the Referee judges our work as “really interesting”, “intriguing”, presenting “interesting conclusions”, “conceptually interesting”, “definitely of interest to the broad readership of Nat. Comm.” We also thank the Referee for the thoughtful comments, which we fully considered in the revised version as follows.

(1) In line 87/88 the authors dub this expectation "generalized translation symmetry", but I am not sure if introducing this terminology really helps the reader in understanding the underlying physics. This remark goes for more of the terminology used/introduced by the authors - for instance the authors distinguish between electronic, atomic and spin magnetization, which is not very useful nor physical, as all magnetization discussed in the paper has the same source: the electron spin. Also the authors introduce for instance "atomic spheres" which are a useful concept in band-structure methodology, but barely a physical quantity.

Reply: We agree with the referee that all types of the magnetization considered in the paper have their origin in the spin magnetization of electrons. However, when addressing different aspects of the magnetism of the system it is very useful to enrich the terminology with terms reflecting the specific features of these aspects. For example, the magnetic structure of the magnetic materials is commonly determined in terms of the directions of the atomic moments. To calculate this quantity we integrate the spin density of all occupied electronic states within atomic spheres. Thus the atomic sphere is a very important concept allowing to straightforwardly determine such a crucial physical quantity as atomic magnetic moment.

On the other hand, we are interested in the energy-resolved magnetization in vacuum at the position of the tip. To help the reader in understanding the paper it is crucial to make these different aspects of the electronic magnetization clearly distinguishable.

We would like to comment on the reference to the generalized translational symmetry. To be precise in our statements, we cannot avoid the use of generalized translation symmetry. A helical structure possesses a counter-intuitive property of equivalence of all atoms. In a usual crystal such an equivalence is a direct consequence of the usual translational symmetry. In a spin helix, the usual translational symmetry is disturbed since the directions of the atomic spins are different. However, if the SOC and, therefore, magnetic anisotropy is negligible the Hamiltonian of the system is invariant with respect to the generalized translations combining the usual translations and pure spin rotations. The group of generalized translations is responsible, for example, for the validity of the generalized Bloch theorem, the possibility to reduce in an exact way the consideration of an infinite magnetic unit cell to the consideration of a small chemical unit cell, the reproducibility of the equivalence of the atoms in the DFT iterations for an incommensurate spin helix.

Following the comments of Referee 3 we include the proof that the magnetization \mathbf{m}_{el} is always collinear to the underlying atomic moment for the ideal helix. In this proof we again need to make reference to the generalized translational symmetry of the ideal helix.

We added information about the origin of the magnetizations in the main text of the paper (L62-71) that should elucidate for the general reader of *Nature Communications* that the terminology used in the paper has the only purpose to make our main results clear and easily comprehensible.

*(2) Another one is the use of "spinor" and "spinor nature" of electronic states. Spin-states are of spinor nature - little controversy about that. The spinor nature of *bands* is *not* probed in the present work (it can be in for instance topological insulators). Apart from this there is no experiment that can avoid probing the spinor nature of magnetization. These kind of embellishments are imprecise and in the end obscure the findings of the paper - which are really interesting.*

Reply: We agree with the referee that is a common knowledge that according to the Dirac's theory the electronic states are four-component spinors. However, if the relativistic effects, in particular the spin-orbit-coupling, are small, they can be neglected. This is done in our paper and the physical equations receive special features. For example, in the consideration of the non-relativistic collinear magnets, the spin-up and spin-down electron states are traditionally described in terms of scalar functions, since the absence of spin mixing makes the use of spinors redundant. In some cases it is even widely accepted to talk about spin-less electrons if the spin-index of the wave function does not play role for the physics discussed.

In our work, it is essential to treat the electronic states as spinors since non-collinear magnetism cannot be described in terms of scalar wave functions. Also in the case of ideal helices the use of spinors in the calculation of the wave functions is mandatory. However, because of the symmetry of the ideal spin helix, the measured direction of the sample magnetization at the tip position is collinear to the direction of the underlying atomic moment, and there is no energy dependence of this direction. In this sense, the specific spin-mixing properties of the electronic states are not observable in the sp-STs experiment. For the distorted helix, the situation is very different.

Here, \mathbf{m}_{el} has an energy dependent direction, making the energy-dependent spin-mixing observable in sp-STs experiment of an ideal spin helix. This allows us to conclude that the sp-STs experiment probes the spinor nature of the electronic states in a rather direct way.

(3) It should be mentioned that the authors do not define the "differential conductance asymmetry" in the main text and only do so in the Methods section and elaborate up on it in the Supplementary Information. The authors connect this asymmetry directly to "spin polarization", which in Fig. 3 is provided in arbitrary units. Again, introducing this kind of terminology and treating it in a vague manner does not help the clarity of the paper. Rather I would expect a discussion in the paper how and under which conditions the energy dependence of the

"differential conductance asymmetry" is related to the energy dependent real-space spin-polarization of the electronic bands.

Reply: We thank the referee for pointing out lacking definitions. We now introduce explicitly the definition of differential conductance asymmetry and its relation to the spin polarization explicitly in the experimental section. We also refer to the generalized Tersoff-Hamann model (PRL 50, 1998(1983), PRL 86, 4132(2001), Science 327, 843(2010)) to discuss the link between dI/dV asymmetry and spin polarization. These reference are now given in the text.

We moved the definition of differential conductance asymmetry from the Methods section into the main text of the paper (L84-95) and added an explanation of the relation between asymmetry and spin polarization (L95-98).

(4) The observations of the authors are compared to the result of band structure calculations. This works well. The interesting conclusion of the authors is that - paraphrasing in my own words - close to the boundaries the local magnetization is non-collinear. This is concluded on basis of the energy dependence of the conductance asymmetry close to the boundary: rather than a change of amplitude (which would imply an energy dependence of a collinear longitudinal polarization), there is an energy dependent change of *phase* of the helix - even if these two alternative interpretations are not directly compared in the paper. (What worries me is that the pictures of e.g. Fig. 2 can be distorted by the asymmetry being high at energies at which the tunneling density of states is very low - rather one would prefer a measure of the asymmetry at which the DOS is high.)

Reply: The referee raises an interesting point. The DOS is high in the energy range -0.6 to -0.1 eV, it smoothly decays to smaller values at more positive energies. This can be taken from the experimental results from Fig S3c (red curves), and also from the plot below. The plot of Fig 2h reveals substantial irregularities, which are related to distortions of the magnetic structure, also in the energy range -0.6 (e.g. at position 6, 7) to -0.1 eV (at positions 2, 6), thus at energies, where the DOS is high.

Reviewer #2 (Remarks to the Author):

We are happy to see that the Referee considers our work as “lifting the study to new level not achieved previously”, presenting “relevant data”, “of interest in the broader solid-state physics community”, “written in a very comprehensible style”, “well organized”, judging that “data are of very high quality and have been analyzed carefully”, inferring that “most claims are substantiated by experiment and theory”. We also thank the Referee for the thoughtful comments, which we fully considered in the revised version as follows.

Detailed answer to Referee 2:

(1) Most claims are substantiated by both experimental data and direct comparison with theory. What I didn't like was the explanation of how the spinor angle phi (presented in Fig. 4 and 5) is obtained. To my opinion the procedure is not properly discussed. To fix this point the authors should consider adding another part to the supplemental material where this fitting procedure is described in detail.

Reply: We follow the referee suggestion and include a section to describe the fitting procedure. In the supplementary material we now include the Supplementary Information 9: fitting of the energy dependent dI/dV asymmetry maps, L112-135. We also included a paragraph detailing how we calculate the spinor angle in the Supplementary Information 10, L156-163.

(2) Furthermore, it is not clearly addressed what the role of the confinement is. Is it the fact that inversion symmetry is broken or is there some electronic interference effect?

Reply: An important result of the transition from an ideal helix to a nm-sized sample is a very strong distortion of the magnetic structure at the interface regions. Because of the broken periodicity of the system, the Fe atoms have different atomic environments and become inequivalent to each other. Under these conditions the regular helical structure cannot stay intact. The strong change in the magnetic structure is closely connected with the change in the electronic structure.

We would like to remark that in our calculations all types of coherence of the wave functions of the electronic states are consequently taken into account. The simplified picture of interference effects of running electronic waves is based on the presence of coherence in the system. We however think that the language of interference effect is not useful in the explanation of properties of electronic states calculated in our paper, where we use full power of quantum mechanical density functional theory. What is important is the consequence of distortions of the magnetic structure close to interface.

Concerning broken inversion symmetry of the samples with different interfaces, the strongest effect is the difference of the distortions at the Co and vacuum interfaces. The effect on the central part is weaker and decreases with increasing size of the sample.

(3) *Another issue that appears potentially confusing to me is the following: In the main part of their manuscript the authors seem to suggest that in collinear magnets or infinite films the electronic magnetization is always aligned with the magnetic quantization axes. However, the non-collinear part of the magnetization density (in collinear systems) has been discussed previously (see, for example, PRL 76, 4420 and Refs. 2-5 therein) and also verified experimentally (PRL 86, 2142). In these studies (some of which shall be cited to properly discuss the scientific background) the non-collinearity originates from band structure effects or from spin mixture driven by spin-orbit coupling.*

Reply: The referee raises an important point, which we address as follows.

(i) The presence of SOC always leads to spin-mixing and therefore to the noncollinearity of the spin magnetization. In ferromagnetic systems this non-collinearity takes the form of the intra-atomic non-collinearity as discussed by Nordström and Singh (PRL 76, 4420). To demonstrate this property they considered Pu with a huge SOC. In our system, the source of the non-collinearity is different, it is the exchange interaction. The SOC is very weak in our system and it is neglected. The non-relativistic calculation for any collinear system can be consequently performed in terms of the pure spin-up and pure spin-down electronic states, that is in terms of totally collinear spin magnetization. On the opposite, all attempts to calculate non-relativistic metallic ferromagnet with any kind of assumed intra-atomic non-collinearity, to our best knowledge, always resulted in totally collinear self-consistent magnetization.

(ii) Bode et al (PRL 86, 2142) report a remarkable experimental property of the bias-dependent variation of the tip magnetization. Such a tip allowed them to extract additional information on the sample studied. In our study, the tip maintains a fixed magnetization orientation along the sample normal in the energy range.

We made the following changes in the manuscript: (a) we refer to the paper by Bode et al. PRL 86, 2142 (b) we prove that in our case the direction of the tip magnetization does not change with bias and therefore we discuss properties related to the sample and caused by the lateral confinement.

(iii) Following recommendation of the referee we added references to the PRL 76, 4420 and PRL 86, 2142, in the main text: L218-L230.

We include Supplementary Information 4 to describe the properties of the tip spin polarization.

We also add (Sandratskii L. M and Guletskii P. G., J. Phys F: Metal Phys.16 L43 (1986); Mryasov et al. J. Phys.: Condensed Matter 3 7683 (1991); Uhl et al. J. Magn. Mater. 103, 314 (1992)) in L135.

(4) *One (probably minor) point:*

Why is the experimentally observed asymmetry (Fig. 1e) so much larger than the calculated spin polarization (Fig. 2c)? Usually it's the other way around.

Reply: We understand that the Referee refers to Figs. 2e and 3c. The quantitative comparison of the theoretical curve Fig. 3c with the experimental asymmetry data of Fig. 2e should be taken with caution. For an individual electronic state the ratio of the projection of the magnetic moment in the vacuum sphere on the direction of the underlying atomic moment, measured in μ_B , to the charge in the sphere often reaches the value of 0.6 and higher. However, there are states where the spin-polarization with respect to the global axis is much smaller or it is even of a different sign. In general, the energy dependence of the relative polarization shows rather sharp features over a narrow energy range. This is often observed for calculated densities of states. Since the experiment never shows a comparable fine structure we performed an averaging procedure by convolution of the calculated relative spin-magnetization density with a Lorentzian function with the width of 0.16 eV. All electronic states with the energy in the corresponding interval enter the procedure with the same weight. This procedure is stable with respect to the main features of the energy dependence of the relative spin polarization. In combination with fundamental symmetry properties it provides a reliable basis for the conclusion derived in the paper.

To allow a quantitative comparison a much more complex calculation of the tunneling current must be performed. This goes beyond the topic of the present study.

Reviewer #3 (Remarks to the Author):

We are happy that the Referee acknowledges that our work “report(s) a surprising change in the non-collinear phase”, and judges that “experimental results are of high quality”, “authors provide a very complete analysis”, “spectacular observation”, “results (are) of incredibly high quality”. We also thank the Referee for the thoughtful comments, which we fully considered in the revised version as follows.

Detailed answer to Referee 3:

(1) Fig. 4 represents the main result of this manuscript: the energy dependence of the non-collinear phase difference between atomic neighbours measured for the Co-Fe-Vacuum case. The spectacular observation of a phase variation in the range between -0.2 eV to 0 eV is correlated to the calculated spinor angle, which shows also variations between -0.2 eV and +0.2 eV. This apparently confirms that the measurements reproduce the spinor nature of the electronic states (as they mention in page 202). However, I never had any doubt of that the sp-STS would reproduce variation in the electronic spin polarization; this is the basis of sp-STS (as said also in line 44).

Reply: We agree that it is common knowledge that the contrast in sp-STS measurements depends on the variation of the spin magnetization. We also use for the analysis of our results the well-accepted formula derived by Wortmann et al. (PRL 86, 4132 (2001)). However, we do not know another study where the energy dependent spin mixing is unambiguously established by a combined experimental-theoretical study. This energy-dependent spin-mixing leads to an energy-dependent magnetization direction, and it is a specific feature of the spinor wave function that cannot be described in any approach dealing with scalar wave functions. We find it important to bring this fundamental connection between the measured property and the spinor nature of the electronic states, leading to spin mixing, to the attention of a general *Nature Communications* reader.

(2) The paper leaves still unsolved the origin of the main claim, namely, that the asymmetric barrier case shows such peculiar energy evolution of the NCM phase. Many questions regarding the origin of such phase change, the energy range where it occurs, and the role of the Co interface remain open.

Reply: The referee raises very important issues.

(i) The asymmetry of the barrier plays a secondary role. The most important statement of very general validity is the following: An irregular non-collinear atomic magnetic structure leads to an energy dependent electronic magnetization. In particular, this leads to the non-collinearity between the electronic magnetization and that of the corresponding atomic moments. We first show that an irregular (non-symmetric) distortion of the magnetic structure results. We demonstrate the energy dependence of the direction of the electronic magnetization.

(ii) The physical picture suggested in our manuscript is fairly complete and well founded. However, to explain the concrete details of the energy dependence of the electronic magnetization direction in vacuum at the tip position is very difficult. The physical origin of the phenomenon lies in the distorted magnetic structure of the sample. The electronic states adopting themselves to the complex potential landscape originated from the irregular magnetic structure and decaying exponentially into vacuum result in the measured energy dependent quantity. At the moment we have some indications based on theory to say that the strong variation of the direction of m_{el} in the vacuum takes place in the energy interval where atomic Fe 3d-up and 3d-down densities tend to compensate each other. But again, the process of the formation of the electronic states in irregular magnetic structures is very complex and, at the moment, we are not ready to make a concrete statement regarding the interplay between the magnetic structure of the sample and details of the energy dependence of the electronic magnetization direction in the vacuum region.

(3) The authors mention in the abstract that "theory reveals that the lateral confinement-induced symmetry breaking is the driving force..." , but this conclusive sentence is not supported in the presented results neither in the discussion. Why confinement produces this effect? I also miss more specific details about the theory employed, that would allow a reader to reproduce the results: which code/functional do the authors use? does theory take into account spin polarization of states several Angstroms above the surface? If so, at which distance?

Reply: An important result of the transition from an ideal helix to a nm-sized sample is a very strong distortion of the magnetic structure at the interface region. Because of the broken periodicity of the system, the Fe atoms have different atomic environments and become inequivalent to each other. Under these conditions the regular helical structure cannot stay intact. The strong change in the magnetic structure is closely connected with the change in the electronic structure.

Concerning the details of the theoretical approach, we used the Augmented Spherical Waves (ASW) method and employed the exchange correlation functional in the form suggested by von Barth and Hedin (von Barth J. Phys. Solid State Phys. 5 1629 (1972)). The spin-polarization of the electronic states in the vacuum region was consequently taken into account. For comparison with the experimental asymmetry signal we considered the vector spin magnetization calculated in the spheres with the center lying 4 Å above the surface.

This information is now added to the Methods section: L287-L290 and L292-L295.

(4) Additionally, pure Fe islands also show variations of NC spin polarization in dI/dV profiles, shown as supplementary Material for the case of pure Fe islands. However, although DFT simulated the vacuum-Fe-vacuum case (also in Sup. material), the authors did not present the calculated energy dependent of spinor components. Can they provide an explanation of this effect on pure Fe islands too?

Reply: We now show in the Supplementary Material the theoretical angle dependence of the direction of the electronic magnetization calculated at the interface Fe atom of the Vac/Fe/Vac sample. The physical origin of the energy dependence is exactly the same as in the case of Fig. 5 shown in the main text.

We added now a Fig. S10, which includes the spinor angle as a function of energy in the case of symmetrical boundaries (Vac/Fe/Vac) and the respective text in Supplementary Information 11: L195-204.

(5) Although the authors present results of incredibly high quality, both theoretical and experimental, the main claim of the manuscript origin of the main observation, non-collinearity between sample and electronic magnetization, is not fully explained. It is also claimed that "It is absent for collinear magnets and ideal spin helix structures." But for pure Fe islands, it is also observed to less extent. I suggest the authors to work out more in detail this part of the manuscript, without which I cannot recommend publication of this manuscript in its current form.

Reply: We now include the proof of the statement that for an ideal spin helix there is no energy-dependent variation of the direction of the electronic magnetization. The pure Fe island does not satisfy the conditions of the ideal infinite helix and consequently, it must possess the energy dependent variation of the direction of the electronic magnetization. We clearly see this in the calculations in the region close to the interface. In the experiment, the growth of larger Fe-b islands is inhibited (Biedermann et al. Phys. Rev. B 73, 165418 (2006)). Our experimental approximation of the ideal spin helix is subject to irregularities caused by the limited size of the island (9 nm lateral extension). In summary, the experiment not only distinguishes the strongly distorted spin helix from the almost ideal one, it also notices finer features that reveal the influence of the lateral confinement on experimental approximation of the ideal spin helix.

We added explanation of the ideal spin helix collinearity between electronic magnetization and magnetic moment in the Supplementary Information 10. L164-L182.

REVIEWERS' COMMENTS:

Reviewer #1 (Remarks to the Author):

The rebuttal of the authors and the resulting adjustments to the manuscript are convincing. The paper now meets all criteria for publication in Nature Communications.

Reviewer #2 (Remarks to the Author):

I read the comments of all referees and the author's responses to these comments. To my opinion the response is adequate and removes the vast majority of ambiguities/questions identified by the referees in the first version of the manuscript. Therefore, I support publication of the manuscript in its current state.